# Effect of Irrigation Dose on Powdery Mildew Incidence and Root Biomass of Sessile Oaks (*Quercus petraea* (Matt.) Liebl.)

**DOI:** 10.3390/plants11091248

**Published:** 2022-05-05

**Authors:** Winicjusz Kasprzyk, Marlena Baranowska, Robert Korzeniewicz, Jolanta Behnke-Borowczyk, Wojciech Kowalkowski

**Affiliations:** 1The State Forests National Forest Holding, Forest District Jawor, Myśliborska 3 Str., 59-400 Jawor, Poland; seeman6@wp.pl; 2Faculty of Forestry and Wood Technology, Poznan University of Life Sciences, Wojska Polskiego 28 Str., 60-637 Poznań, Poland; robert.korzeniewicz@up.poznan.pl (R.K.); jbehnke@up.poznan.pl (J.B.-B.); wojciech.kowalkowski@up.poznan.pl (W.K.)

**Keywords:** *Erysiphe alphitoides*, the shoot-biomass amount, Compu Eye, Leaf & Symptom Area

## Abstract

The sessile oak is one of the most significant forest tree species in Europe. This species is vulnerable to various stresses, among which drought and powdery mildew have been the most serious threats. The aim of this study was to determine the influence of irrigation levels (overhead sprinklers) on the damage caused by powdery mildew to *Quercus petraea* growing in a nursery setting. Four irrigation rates were used: 100%, 75%, 50% and 25% of the full rate. The area of the leaves was measured and the ratio between the dry mass of the roots and the dry mass of the entire plant was calculated after the growing season in years’ 2015 and 2016. Limiting the total amount of water provided to a level between 53.6 mm × m^−2^ and 83.6 mm × m^−2^, particularly in the months when total precipitation was low (VII and VIII 2015), a supplemental irrigation rate between 3 and 9 mm × m^−2^ resulted in a lower severity of oak powdery mildew on leaves and lead to a favorable allocation of the biomass of the sessile oak seedlings to the root system. The severity of infection on oak leaf blades was lower when irrigation rates were reduced. The greatest mean degree of infestation in 2015 was noted in the 100% irrigation rate (14.6%), 75% (6.25%), 50% (4.35%) and 25% (5.47%). In 2016, there was no significant difference between the mean area of leaves infected by powdery mildew depending on the applied irrigation rate. The shoot-root biomass rate showed greater variation under limited irrigation rates. Controlling the irrigation rate can become an effective component of integrated protection strategies against this pathogen.

## 1. Introduction

The sessile oak (*Quercus petraea* (Mattuschka) Liebl.) is one of the most significant economic and ecologic forest tree species in Europe. This species is vulnerable to various abiotic stresses, among which drought has been considered to be the most serious threat [1]. Apart from the negative influence of abiotic factors on oak trees, there are also additional unfavorable biotic factors that affect the trees [2]. Powdery mildew caused by *Erysiphe alphitoides* (Griffon and Maubl.) U. Braun and S. Takam. is one of the most dangerous diseases of oaks. *Erysiphe alphitoides* is a non-indigenous fungal species from North America that was first reported in Europe in 1907 [3] and was first studied in Poland in 1909 [4].

Other species of *Erysiphe* spp. that occur on oak leaves include: *E. abbreviata* (Peck) U. Braun and S. Takam., *E. calocladophora* (G.F. Atk.) U. Braun and S. Takam., *E. extensa* (Cooke and Peck) U. Braun and S. Takam., *E. gracilis* R.Y. Zheng and G.Q. Chen, *E. hypophylla* (Nevod.) U. Braun and Cunningt, *E. quercicola* S. Takam. and U. Braun and *E. polygoni* D. C. [5,6,7]. The use of molecular biology techniques allowed us to state that they can occur in Europe in four haplotypes of the pathogen. Three of them are from *E. alphitoides* (100% ITS compliance), *E. hypophylla* (99.4% ITS agreement) and *Phyllactinia guttata* (Wallr.) Lév. (97.64% ITS agreement). A fourth, common in Europe, has 100% ITS compatibility with *E. quercicola* found in *Quercus* species in Asia and several tropical mildew species including *Oidium heveae* B.A. Steinm [8]. Sucharzewska [9] reported that in Poland, there are two species of powdery mildew: *E. alphitoides* and *E. hypophylla*. *Erysiphe hypophylla* is more numerous in the western part of Poland. These results are also confirmed by Behnke-Borowczyk and Baranowska-Wasilewska [10] and Roszak et al. [11], because during the study of powdery mildew affecting oaks from western Poland, they confirmed the presence of only *E. alphitoides.*

*Erysiphe alphitoides* causes a disease of the leaves and affects the non-lignified tissues of oak trees of a range of plants including *Quercus robur* L., *Q. conferta* (Kit.) Vuk., *Q. pubescens* Willd., *Q. cerris* L. and *Q. pyrenaica* Willd. [12]. It has been identified also on the leaves of *Fagus sylvatica* L., *Castanea sativa* Mill., other host plants such as on the *Eucalyptus gunnii* Hook. f. London [13]. Powdery mildew is an obligate parasite that attacks oaks of all age classes [8,14], resulting in exponentially increasing damage, particularly in young seedlings in the early phases of ontogeny [15,16], but only young developing leaves are susceptible of the disease [17]. Increased *E. alphitoides* prevalence in spring is associated with higher oak mildew severity in autumn [18]. Annually occurring infections can lead to necrosis and eventually death of young trees [19]. Non-lignification of the affected shoots may enable recurring infections, which occur in summer and concern mainly fresh and regenerating leaves [2,20]. Oak powdery mildew reduces the uptake of CO_2_ (net), hinders stomatal conductance and lowers the nitrogen content of leaves [15]. Direct exposure to sun light as well as a warmer and dryer microclimate (e.g., conditions in gaps of a tree stand) both enhance spread of oak powdery mildew [21,22]. Selochnik et al. [23] mentioned an optimum disease development for mean temperature in June around 20 °C. Research pointed to the detrimental effects of rain on powdery mildew fungi, by washing off spores and damaging mycelium at the leaf surface [24]. Sivapalan [25] showed that *E. alphitoides* for which conidia were able to germinate as well in water as on oak leaves, the ability of the fungus to establish a parasitic relationship with the host strongly decreased with the duration in water.

Due to the economic significance of oaks, powdery mildew is of interest to scientists. Research has focused, for example, on understanding the influence of abiotic factors and climate change on promoting spread of the pathogen [12,15,26,27], as well as on the possibilities of exploiting biological methods to reduce the occurrence of oak powdery mildew [28,29]. Lately, it has been reported that the failure of natural regeneration of pedunculate oak trees in Europe attributable to infestation by oak powdery mildew is linked to factors such as animal pressure, changes in land use and the diminishing levels of ground water [30]. However, an interesting example is given by Demeter et al. [31], who reported the failure of natural regeneration of sessile oaks, despite the lack of animal pressure, sufficient accessibility to water and no changes in land use, in a region where prior to the appearance of oak powdery mildew, there was successful widespread natural regeneration of oak trees [32]. Every year, the native tree species range of tree stands in Europe, including oak trees, increases. It is associated with a more extensive culture of oak trees in nurseries, and this, in turn, results in a greater risk of outbreaks of powdery mildew [11]. In the nurseries, which are producing seedlings of trees, several irrigation and methods are available and used. Irrigation can reduce leaching, time of work, use of fertilizer and foliage diseases. Subirrigation systems are commonly used by nursery plant producers. Drip irrigation has been used in areas where water is scarce. Drip irrigation uses up to 70% less water as compared to a flood irrigation system. However, it is expensive compared to other irrigation methods [33]. Drip irrigation was effective in reducing pathogens such as *Colletotrichum acutatum* (J. H. Simmonds) that cause anthroids in strawberry production [34]. Overhead irrigation is the most common system in outdoor nursery areas. Overhead irrigation is water inefficient (as much as 80%). Major irrigation methods chosen for field nursery irrigation system are overhead sprinklers and microirrigation. The choice of these methods, again, depends on crop, water type and quality of plants [33].

In this context, it is of particular interest to determine whether standard practice in nurseries, i.e., irrigation, can be modified to limit the occurrence of powdery mildew in sessile oaks. Any modifications to irrigation schedules should be implemented in a manner that does not distort the normal shoot-root ratio and will not lead to lasting physiological changes in plants. Hence, the aim of the current study was to determine the influence of the irrigation rate provided for the sessile oaks (in a field nursery) on the damage caused by powdery mildew. It is hypothesized that different irrigation rates are likely to impact the severity of the sessile oak leaf infestation caused by powdery mildew. Specifically, we predict that under a lower irrigation rate (i) the severity of infection of the leaf blade of the oak will be reduced and (ii) the biomass ratio of shoots and roots will be greater.

## 2. Results

Analysis of the variability in soil moisture content in 2015 showed statistically significant differences between variants, blocks and variability over time. The 25% variant was clearly less wet than the other variants on the first two measurement dates. As a result of the deepening drought and the increase in temperatures in the third measuring period, a reduction in humidity was also observed in the 50 and 75% variants (Table 1). A consistent reduction in the general level of soil moisture by weight was also observed, and deepened with time. The results of measurements of moisture content by weight and volume carried out in 2016 did not show statistically significant differences. The correlation analysis showed the existence of a statistically significant relationship between the weight and volume moisture, which was the basis for the calculations that allowed derivation of the regression equation (Table 1).

ANOVA revealed that depending on the adopted irrigation rate in 2015, the area of oak leaves affected by powdery mildew differed significantly (F = 11.71; *p* < 0.001). In 2016, there was no statistically significant difference between the mean area of leaves infected by powdery mildew depending on the applied irrigation rate (F = 2.04; *p* = 0.115) (Figure 1a,b). The greatest mean degree of infestation in 2015 was noted in the case of 100% irrigation rate (14.6%) and with α = 0.05, differed significantly from the irrigation rate (75–6.25%; 50–4.35%; 25–5.47%; Figure 1 and Figure 2). In 2016, the mean area of leaves infected, regardless of the irrigation variant, fluctuated between 9.33% and 12.54%.

The ratio of the dry root mass to dry seedling mass of the sessile oak (F = 6.85; *p* = 0.0004) differed significantly between the different irrigation rates in 2015, and also in 2016 (F = 3.67; *p* = 0.016). The greatest mean value of the ratio in 2015 was observed in the case of oaks which had been irrigated with 50% of the irrigation rate (0.64), and the second highest value occurred at 75% of the irrigation rate (0.65). These results differed significantly from the mean for 100% of the irrigation rate (0.54). There were no statistically significant differences in the mean ratio of the dry root mass to dry seedling mass in the case of oaks irrigated with 25% of the irrigation rate (0.59) and the other means (Figure 3a,b). However, in 2016, the highest value of this ratio was observed in the oaks irrigated with 25% of the irrigation rate (0.54) and this differed significantly from the value for the oaks irrigated with 75% of the irrigation rate (0.46). The mean ratio did not differ from others in the case of the oaks irrigated with 50% (0.5) and 100% of the irrigation rate (100%) (Figure 3a,b).

## 3. Discussion

The results reported herein show that it is possible to influence the degree of powdery mildew infestation, which occurs on the leaves of the sessile oak, by controlling irrigation. The experiment confirms Sharma et al. [35] findings that higher irrigation rate is conducive to the development of the pathogen, but the differences were significant only during the first year. Sharma et al. [35] carried out research to evaluate the effect of irrigation schedules on the incidence of powdery mildew disease (*E. graminis tritici* E. Marchal) in wheat (*Triticum aestivum* L. emend. Fiori and Paol.).

A combination of favorable temperatures and optimal humidity are vital for the survival and transmission of oak powdery mildew [16]. Pap et al. [16] claim that other elements which contribute to the effectiveness of powdery mildew infections are either excess or lack of nutrients and water in the soil, as well as access to space and light. Optimal conditions contribute to the acceleration of ontogenesis, which decreases the severity of powdery mildew on leaves and, at the same time, limits the negative outcomes of oak powdery mildew infection [16]. The optimum for growth of oak powdery mildew is at 25 °C and 96% relative humidity [16,36]. Light had a significant impact on the growth of mycelium of oak powdery mildew that reached the maximum value in full light [16]. Moreover, it has been shown that the more wetted the leaves, the greater the impact of powdery mildew on the plant [37]. The mycelium of oak powdery mildew appears to be hydrophilic [38]. Despite the existing difference in the content of nutrients in the soil, no differences in powdery mildew of oak were noted between the same variants of the experiment, which were located in different blocks of the experiment. In view of all the observations listed above, controlling the irrigation rate would appear to be of key significance in the protection of oak trees against oak powdery mildew in nurseries, and this conclusion is supported by the results of the experiment in 2015.

Water is crucial for the growth and development of plants. A long-lasting period with a lack of sufficient water inevitably leads to dehydration and can lead to the death of a plant. It can also increase susceptibility to infectious diseases [39]. In this context, scientists have been intrigued by the influence of oak powdery mildew on changes in stomatal conductance and other consequences of infection by this fungus [40,41]. Oak powdery mildew has been shown to increase transpiration of the infected leaves. Another element which evoked interest was the impact of the pathogen, which affects only leaves, on water consumption of the entire plant [15]. However, the precise irrigation rate that would impede the development of the disease in nurseries, without disturbing growth and development of seedlings, has not been determined. Taking into account the larger amounts of water in 2016, which came from precipitation and lower air temperatures, it can be concluded that the excess of water may contribute to the intensification of infection from powdery oak mildew.

In 2015 and 2016, weather conditions may have influenced the existence of differences between soil moisture, the amount of leaf infestation by powdery mildew of oak and the dry mass of oak roots. The year 2015 was drier compared to 2016, which confirms the existence of differences in soil moisture in 2015 and no differences in 2016. Lower powdery mildew infection of oak may result from the presence of higher current air temperatures in 2015 and lower rainfall. A similar relationship was observed by Markovic et al. [42], who, at a lower air temperature of 17–21 °C and air humidity of 85–100%, recorded a very high rate of seedling infection.

All powdery mildew species can germinate and get infected in the absence of water [43]. Sucharzewska [9] informs that the greatest degree of powdery mildew infection was noted at times when the growing season had been very warm and dry. According to Marçais et al. [18] it is also possible that a microclimate close to the ground, with higher humidity, may provide good conditions for *E. quercicola* development. The results of the current study indicate that oak powdery mildew infected trees which were irrigated by a range of irrigation rates; however, the degree of leaf infection was less severe under lower irrigation rates, but only in years when total precipitation was lower. Thus, it is possible to decrease the infection of the oak trees in nurseries by limiting and controlling the irrigation rate. As a result, it is advisable to use irrigation rates in the range between 53.6 mm/m^2^ and 83.6 mm/m^2^ in the July–August period.

The mycelium of oak powdery mildew mainly covers leaves [8]; hence, it is possible to use the leaves covered by fungal hyphae in order to assess the severity of the infection [14]. Oak powdery mildews infect young developing leaves in spring and summer [19]. In Europe, the white mycelium of oak powdery mildew on leaves and shoots often appears in early spring [15]. The secondary infections occur during the summer and concern mainly the young foliage regenerating after insect defoliation [2]. While analyzing the severity of infection on oak leaf blades in post-flood trees stands (leaves in full isolation), concluded that the maximal infection by powdery mildew did not exceed 22%. Our results concur because even with the highest irrigation rate, the infection did not exceed 22%. These findings indicate that the higher irrigation rates in nurseries increase in degree of the leaf blade infection by *Erysiphe* spp.

It is worth emphasizing that numerous studies on oak powdery mildew have focused on the interactions between the pathogen and pedunculate oak trees [15,16,37,44,45,46], and there are fewer publications on the interactions of this fungus and sessile oak trees [18,19,47,48], most likely because sessile oaks occur less frequently (cover smaller areas) [48]. Nevertheless, the latter species is a valuable component of deciduous tree stands, and both oak species can freely interbreed [49]. Given that the sessile oak is more resilient to droughts than the pedunculate oak [50], it is probable that the species will gain in significance in the future, particularly in view of climate change. Hence, it is important to devote more attention to the study of the interactions between stress factors, such as oak powdery mildew, and the sessile oak.

The variation of irrigation rate that was applied in the current work influenced also the rate of development of the root-mass. In the dry areas, seedlings increased water consumption and invested more in root system development. This later allowed the trees to draw water more effectively from deeper layers of the soil, where greater reservoirs of water reside especially in dry seasons [51]. That is why using lower irrigation rates stimulated oaks to invest more in development of the root biomass. Similar dependence has been reported by Gieger and Thomas [52], Thomas and Gausling [53] and Sustani et al. [54]. However, Broadmeadow [55] noticed a clear influence of optimal water conditions on the shaping of unfavorable shoot-root rate in the sessile oak caused by the excessive increase of the aerial component of the trees. The relationship between the shoot of Scots pine and the shoot of ash and root biomass was affected by irrigation. These plants tended towards increased shoot allocation. However, in oak, there were large differences in the shoot-root relationship. These are difficult to explain in terms of irrigated effects, which is also confirmed by reports by Broadmeadow et al. [55].

## 4. Materials and Methods

### 4.1. Experiment Design

The experiment focused on sessile oaks which were grown from seeds collected from a seed-tree stand (stand age–159 years) in subdivision 315d in Legnica Forest District (Poland, WGS 84–51.3016 E:16.0833). The experiment was established in the Muchów Forest Nursery in subdivision 194f in Jawor Forest District (Poland, WGS 84–51.0076 E:16.0187)-open field nursery.

The seeds were planted by hand in an open field nursery (0.606 kg acorns/m^2^) on the 6 May 2015 in 4 rows, and were then covered with sand and protected with nonwoven fabric (spacing between seeds was 5 cm and between rows was 33.3 cm, Figure 4d). During the experiment (in 2015 and in 2016), no protective measures were undertaken. A randomized complete block design, with 4 blocks, was used in the experiment. Each plot was 2 m long and 1.5 m wide (about 200 oaks grew on each plot; Figure 4a–d.). Four irrigations rates were used: (1) a fully-calculated rate–100% (control group–12 mm/m^2^; reference evapotranspiration-ETo), (2) 75% of the full rate (9 mm/m^2^), (3) 50% of the full rate (6 mm/m^2^), (4) 25% of the full rate (3 mm/m^2^) (Figure 4c). Density of the trees in each experimental unit was uniform. Plants were surface irrigation-overhead sprinklers method (Figure 4a,b). The atmospheric conditions were the same in all places of the experiment (Table 2 and Figure 5).Variations of the irrigation rate were established based on the document: “Guidelines for irrigating forest nurseries on open areas” [56]. In order to achieve initial stable and balanced growth in the period following sowing of the acorns, the irrigation rate on all plots was identical and as advised [56]. A necessary irrigation rate for the so-called 2nd period of irrigation of a one-year-old material was determined on the basis of Smorowski’s method described in the Guidelines document [56].

The rate of easily accessible water in the soil was verified at 7.7% of the soil volume. Average daily water consumption for evapotranspiration (similar to ETo) was assumed to be 2.3 mm. High annual precipitation exceeding 610 mm [56]. The preferable depth for dampening the soil was established at 15 cm in both years of the experiment. The net irrigation dose (d) was calculated according to the formula:d = 0.1 × Wd × h [mm],
where Wd—water content readily available in % of soil volume; h—the desired depth of soil moistening. The gross irrigation dose (d) was calculated according to the formula:D = d/k_e_ [mm],
where d/k_e_—technical efficiency factor of irrigation (0.85) [23]. The frequency of watering (T) was calculated by the formula:T = d/E,
where d—the net irrigation dose, E—daily water consumption for evapotranspiration [23].

The calculated net dose of irrigation was 11.55 mm (13.58 mm gross); however, the calculated frequency for sprinkling irrigation was 5 days. In order to determine the sprinkler flow rate per 1 h, a pluviometer from a Davis Vantage Pro weather station was used, which was positioned centrally on the experimental fields (before seeding). The achieved mean value of the flow rate per 1 sprinkler was 3.5 mm × m^−2^ × h^−1^. RainBird EWH 14070 sprinklers were used in the experiment (Figure 4a,b). The adopted methods permitted careful control of the irrigation rate for each variant by controlling sprinkler run time. The sprinkler was turned off after a set period during which a precise irrigation rate for the particular variant was delivered.

Precipitation reaching 3 mm, which occurred between each artificial irrigation, was also taken into consideration. Plants in the 100% dose variant were watered between 5:00 a.m. and 9:00 a.m., 75% from 5:00 a.m. to 8:00 a.m., 50% from 5:00 a.m. to 7:00 a.m. and 25% from 5:00 a.m. to 6:00 a.m. If precipitation occurred, then the next irrigation rate was rectified according to the extent of the precipitation [56]. In cases when the sum of precipitation in the period from the earlier irrigation reached the approximated value to 1 full irrigation rate, then the date of the next irrigation was postponed by another 5 days. This situation only happened twice: on 10 July 2015 and 19 July 2015, when above 3 mm of rain fell the day before. The total quantity of water from the sprinklers and from the precipitation is presented in Table 1. Meteorological data during the time of the research are given in Figure 4. Average data are derived from the closest meteorological station in Wojcieszów Górny (Poland, WGS 84–50.9516 E:15.9214.), while the local data come from the Davis Vantage Pro meteorological station, located in the Muchów Forest Nursery. We have not noticed the influence of the seasons on the development of powdery mildew of oak.

### 4.2. Soil Moisture

Chemical analyses of soil were commissioned to the Seed Testing Station of the National Forests Research and Implementation Centre in Bedoń (report no 7 of 25 March 2014—internal data, unpublished), where in two soil samples provided for analyses the following parameters were determined: pH in KCl using the electrochemical technique, contents of phosphorus and potassium according to Egner-Rhiem by inductively coupled plasma atomic emission spectrometry [57], magnesium content according to Schachtschabel by inductively coupled plasma atomic emission spectrometry [58], nitrogen content by the direct method using the TruSpec CHNS apparatus [57] organic carbon content by the modified Tiurin method [57], while contents of N-NO_3_ and N-NH_4_ by the electrochemical method following extraction in 0.03 N acetic acid [57]. The results of soil analyses are given in Table 3.

In order to confirm or exclude the influence of the irrigation rate on the soil moisture content by weight and volume, three measurements of soil moisture content by the weight method were carried out on each experimental plot in 2015 and one measurement of soil moisture content with simultaneous measurement of volumetric moisture in 2016. Moisture measurements were made after 4 h from the end of irrigation. In 2015, 64 samples were taken with an Egner cane at a depth of between 10 and 15 cm. Before drying, each sample was thoroughly mixed and a cohesive sample, weighing 50 g, was separated from it by pouring the soil into a ceramic container with a constant weight placed on a weighing pan. Then, the samples were dried at a temperature of 105 °C until a constant weight was obtained, which lasted about 4 h. After completion of drying, the soil samples together with the containers were weighed to an accuracy of 0.001 g.

The weight moisture content of each soil sample was calculated according to formula:$$W=\frac{\mathrm{mmt}-\mathrm{mst}}{\mathrm{mst}-\mathrm{mt}}*\;100\;[\mathrm{mm}],$$
where mt—weight of the container mmt—weight of the vessel with moist soil mst—weight of the vessel with dry soil [59]. The HH2 Meter was used to measure volumetric moisture of the soil in 2016 by Delta T with the ECHO EC-5 sensor by Decagon Devices, which allows to obtain a result in percentage by volume. Volumetric moisture measurements were performed in the places where soil samples were taken to determine the weight moisture content, thus facilitating determination of the relationship between both types of moisture and enabling the derivation of the regression equation.

The data were analyzed statistically implementing a two-factor model without interactions. the data expressed as percentages were subjected to the Bliss transformation.

Additionally, for soil moisture by weight, in 2015, an analysis of variance was performed in a system with repeated measurements in order to determine changes in soil moisture over time. The sphericality of the data was confirmed by the Mauchley test. For weight and volume moisture, in 2016, a linear correlation analysis was performed yielding a correlation coefficient R, determination coefficient R^2^ and estimation of the standard error of the assessment, as well as the coefficients of the regression equation. Statistical analyses were performed using the Statistica 6.0 software (StatSoft) [60].

### 4.3. The Weighing of Dry Mass Seedlings and Calculate Leaves Area

The weighing of dry mass seedlings was conducted after the growing season in years’ 2015 and 2016 (in September-late summer). The leaves, which were collected in order to measure the area of the leaves, were collected before digging out the plants. Leaves were taken randomly from each part of the plant (apical, median, basal). Five trees from each variant of the experiment and from each block in each year of the experiment were extracted (in total 160). From each tree, the leaves were collected for further research (in total 160 pieces). The leaves were scanned in order to calculate their area and, in order to do that, a scanner with a resolution of 300 dpi was used, together with Compu Eye, Leaf & Symptom Area software, which was used to calculate the area of the leaves [61]. The infestation of leaf blades by powdery mildew was expressed as the rate of the affected part (showing etiological symptoms) to the non-infected part (without etiological symptoms) expressed in % (degree of infestation—the severity of the disease, Figure 6). The dry mass [g] of particular organs was weighed after the samples were dried in a thermostatic cabinet (POL-EKO-Aparatura, typ ST 1200 B60 photoperiod) at the temperature of 60 °C. The ratio between dry mass of roots and dry mass of the entire plant was established [g × g^−1^]. Measurements were made on an annual basis due to the invasiveness of the applied methodology, which was dependent on seedlings being extracted from their beds.

Based on previous research by Behnke-Borowczyk and Baranowska-Wasilewska (2017) and Roszak et al. (2019) it was considered that powdery mildew of oak caused by *E. alphitoides* will dominate in the Muchów Forest Nursery.

### 4.4. Statistical Analysis

The Shapiro–Wilk test was used to assess whether data conformed to the Gaussian distribution. In the cases when a normal distribution could not be confirmed, the data were Box-Cox transformation, and after meeting the assumptions of homogeneity of variance, the Levene’s test was applied. When the null hypothesis was rejected (implying no differences between mean values), Tukey’s range test for equal sample size was implemented post-hoc.

## 5. Conclusions

Variation in the rate of irrigation influenced the severity of oak powdery mildew infections on the leaves of sessile oak trees.

The shoot-root biomass rate was greater under limited irrigation rates.

Limiting the total rate of water in July and August (summer) to a level between 53.6 mm × m^−2^ and 83.6 mm × m^−2^, particularly in the months when total precipitation is low, and with a supplemental irrigation rate of between 3 and 9 mm × m^−2^ results in lower severity of oak powdery mildew on leaves. Concurrently, it leads to a favorable allocation of the biomass of the sessile oak seedlings to the root system, which lasts for at least 2 years.

Given the assumptions of the Green Deal, and the implementation of regulations concerning plant protection products, i.e., pesticides [62], it is crucial to conduct further research into the significance of irrigation rates in regulating, and particularly limiting, the severity of oak powdery mildew. It is plausible that controlling the irrigation rate can become an element of the integrated protection method against the pathogen.

Controlling spraying doses (reducing irrigation) can be an effective way to reduce the occurrence of powdery oak on seedlings produced in tunnels and greenhouses; therefore, research should be continued, especially under these conditions.

## Figures and Tables

**Figure 1 plants-11-01248-f001:**
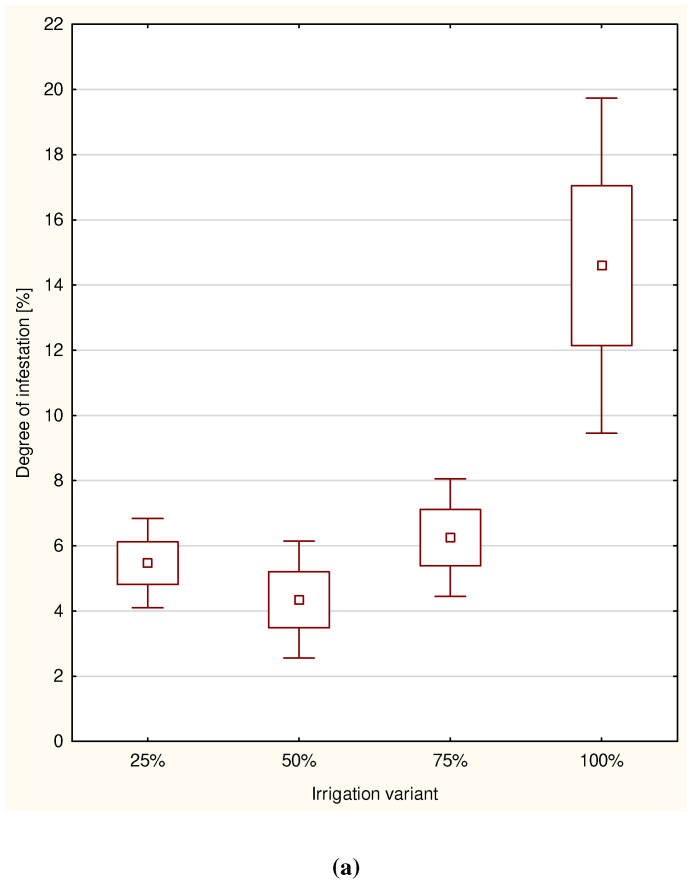
The mean degree of infested leaves of the sessile oak in 2015 (**a**) and in 2016 (**b**) for each variant including standard errors and confidence intervals.

**Figure 2 plants-11-01248-f002:**
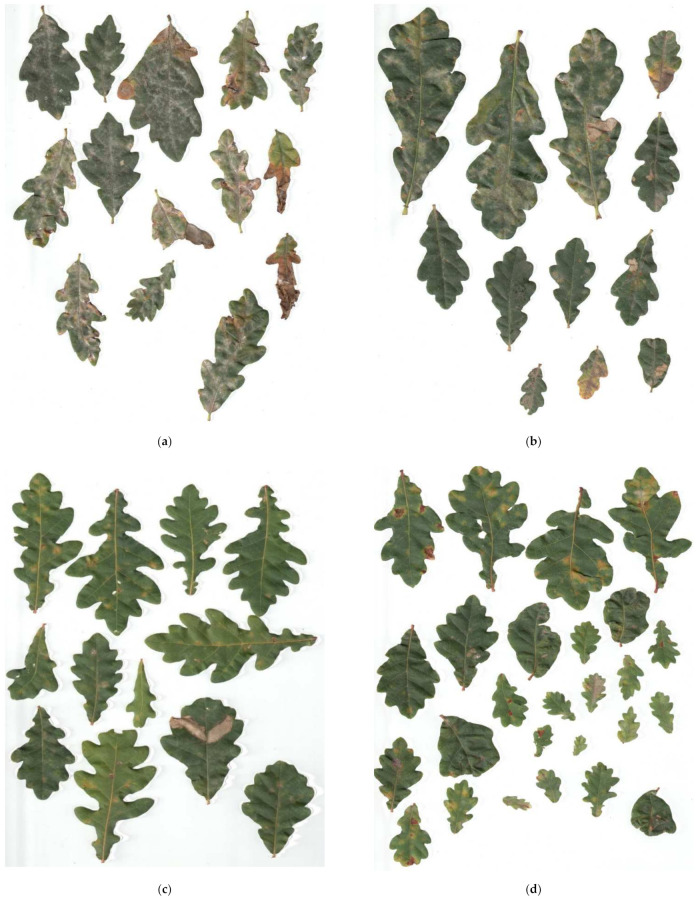
An example of scans of oak leaves taken from plants growing in block A in 2015, infected with powdery mildew of oak, made with a scanner with a resolution of 300 dpi. (**a**) Full dose (100%), (**b**) 75% of the dose, (**c**) 50% of the dose, (**d**) 25% of the dose.

**Figure 3 plants-11-01248-f003:**
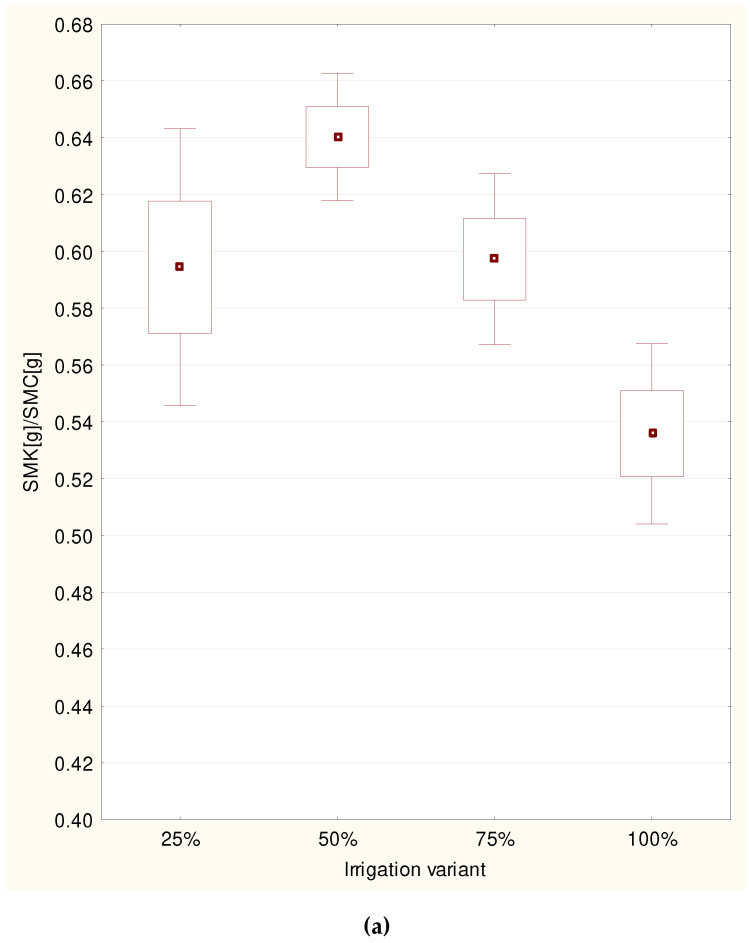
Mean rate of dry root mass (SMK) to dry plant mass (SMC) of the sessile oak in years’ 2015 (**a**) and 2016 (**b**) for each variant including standard errors and confidence intervals.

**Figure 4 plants-11-01248-f004:**
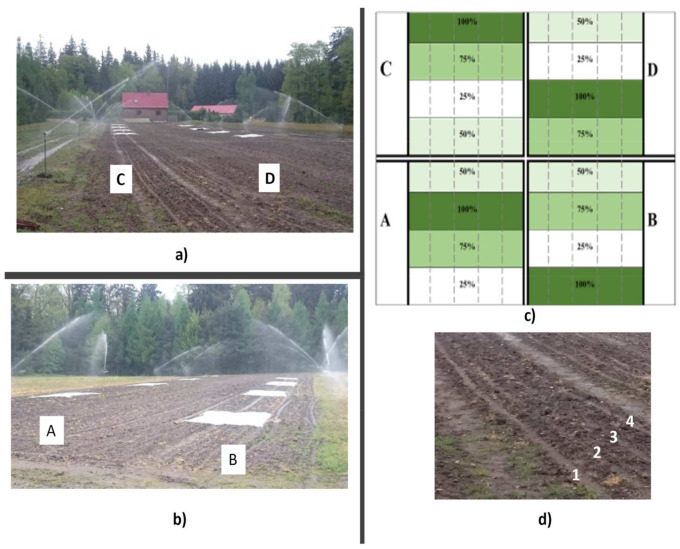
The experimental set-up divided into variants of the experiment in Muchów Forest Nursery; (**a**) Block C and D and working sprinklers, (**b**) block A and B and working sprinklers, (**c**) experiment design; letters stand for block names; 100%-full rate (control), 75% of the rate, 50% of the rate and 25% of the rate; the dashed line marks the rows where the oak was planted; the desired variant of the experiment is marked with a separate color, (**d**) rows in which acorns were planted; the numbers (1–4) are rows.

**Figure 5 plants-11-01248-f005:**
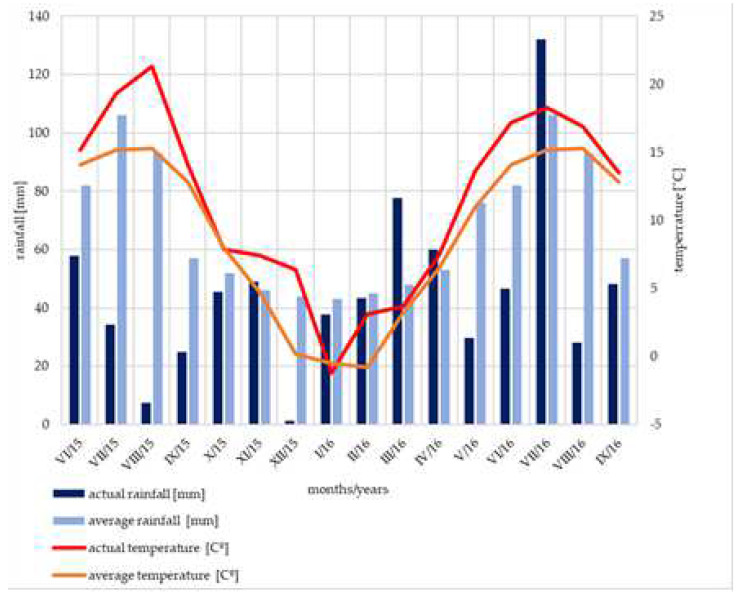
The average rainfall [mm] and temperature [°C] and actual rainfall [mm] and temperature [°C] in Muchów Forest Nursery, Poland (I—January; II—February, III—March etc.).

**Figure 6 plants-11-01248-f006:**
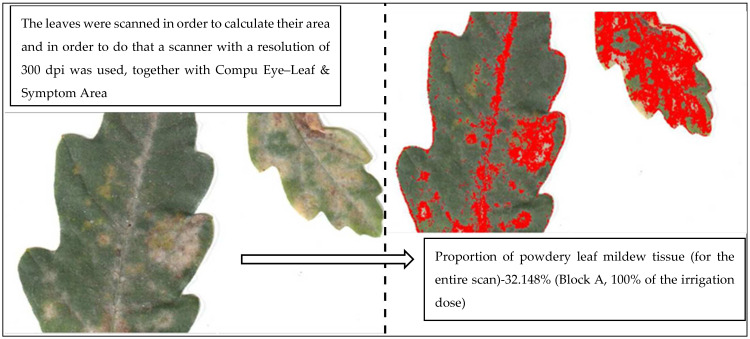
Determine the degree of infestation of scanned leaves with use Compu Eye, Leaf & Symptom Area.

**Table 1 plants-11-01248-t001:** Average values of soil moisture by weight (x, confidence intervals (CI), standard deviations (SD) and coefficients of variation (V) for each variants of experiment in 2015 and in 2016. The same letters (a, b) next to the means mean that the means do not differ significantly (Tukey’s test; α = 0.05).

Date of Measurement	Variant	x	CI −95.0%	CI +95.0%	SD	V
6 July 2015	25%	16.80 (a)	13.36	20.24	6.45	38.39%
	50%	19.87 (b)	17.09	22.64	5.21	26.22%
	75%	19.45 (b)	16.84	22.05	4.88	25.09%
	100%	20.51 (b)	17.26	23.76	6.10	29.74%
28 July 2015	25%	16.80 (a)	12.62	20.97	7.83	46.60%
	50%	20.40 (b)	17.51	23.28	5.41	26.51%
	75%	19.08 (b)	16.91	21.25	4.07	21.33%
	100%	22.03 (b)	18.42	25.65	6.78	30.78%
6 August 2015	25%	15.72 (a)	12.24	19.20	6.53	41.54%
	50%	17.91 (ab)	15.55	20.28	4.44	24.79%
	75%	16.94 (ab)	14.65	19.22	4.28	25.27%
	100%	19.22 (b)	16.16	22.28	5.74	29.86%
2016	25%	18.22 (a)	5.28	31.16	8.13	44.62%
	50%	15.74 (a)	12.16	19.31	2.25	14.29%
	75%	19.17 (a)	8.23	30.11	6.87	35.84%
	100%	18.07 (a)	7.57	28.56	6.60	36.52%

**Table 2 plants-11-01248-t002:** Total quantity of precipitation and irrigation rate for each variant and irrigation dates in years’ 2015 and 2016, VII—July, VIII—August—summer in Poland.

Variant	Total Amount of Precipitation [mm] VII 2015	Dose of Irrigation [mm]				Total Amount of Precipitation [mm] VIII 2015	Dose of Irrigation [mm]	Total Amount of Precipitation [mm] VII 2016	Dose of Irrigation [mm]		Total Amount of Precipitation [mm] VIII 2016	Dose of Irrigation [mm]	
		6 July 2015	19 July 2015	24 July 2015	29 July 2015		5 August 2015		13 July 2016	27 July 2016		16 August 2016	31 August 2016
25%	34.2	3.0	0.0	3.0	3.0	7.4	3.0	132	3.5	3.5	28.2	3.5	3.5
50%		6.0	3.0	6.0	6.0		6.0		7	7		7	7
75%		9.0	6.0	9.0	9.0		9.0		10.5	10.5		10.5	10.5
100%		12.0	9.0	12.0	12.0		12.0		14	14		14	14

**Table 3 plants-11-01248-t003:** Results of chemical analyses of soils.

Parameters	Blocks	
	A and B	C and D
pH in H_2_O	5.8	6.3
pH in KCl	4.6	5.5
N [%]	0.16	0.30
C [%]	2.65	6.58
C/N	16.30	22.00
P_2_O_5_ [mg/100 g]	5.78	9.9
Ca [mg/100 g]	113	326
K [mg/100 g]	9.4	26.7
Mg [mg/100 g]	4.8	15
Na [mg/100 g]	0.32	0.4

## Data Availability

No new data were created or analyzed in this study. Data sharing is not applicable to this article.

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
