# Peer review of "Effect of Irrigation Dose on Powdery Mildew Incidence and Root Biomass of Sessile Oaks (*Quercus petraea* (Matt.) Liebl.)"

_plants, 2022, doi:10.3390/plants11091248_

Round 1

Reviewer 1 Report

This appears to be an interesting and thorough study. I am not in a position to evaluate the quality of statistical analyses, but I did not detect anything wrong with them. 

Suggest changing the title - it is very difficult to read and may be grammatically incorrect
Suggestion:
Effect of irrigation dose on powdery mildew incidence and root biomass of sessile oaks

Abstract 
variation in the extent of irrigation -> irrigation levels

You should state the type of irrigation in the abstract

30-38 I think it is very important to state that E. alphitoides is not the only Erysiphe species to commonly attack oaks, nor is Erysiphe the only PM genus to attack oaks. Then you need to explicitly state if the common term "oak powdery mildew" is referring only to E. alphitoides within the manuscript or all members of the Erysiphaceae that parasitize oaks. Finally, a sentence somewhere in the manuscript that states whether you believe or took any measures to determine whether the symptoms and signs observed in your study come solely from E. alphitoides. 

38 assimilation apparatus -> this term is not familiar to me. This may be my own deficiency, but consider finding a more widely-used synonym

48 claim -> this suggests to the reader you may not agree with the finding -> found
furthermore, this fact sheet is not a suitable citation - use the reference(s) that the fact sheet cites, not the fact sheet itself 

53 "dry" -> replace with something more specific (ie they require humidity but not precipitation)

55 see previous comment about citing factsheet

56 the oak -> either "the sessile oak" or "oaks"

61 the oak -> oak

63 all -> al.

70-75 you should mention irrigation methods in addition to irrigation rates, even if this isn't covered by the study

140  you must address the discrepancy in results between years 2015 and 2016 in the discussion. An entire paragraph would not be too long. 

200 surface irrigated -> flooding or sprinklers? the delivery method is very important for a powdery mildew study

200 was the density of the trees in each experimental unit uniform?

Figure 3 caption replace dose with rate or level

Author Response

Thank you very much for your review. We revised manuscript and we added photos. We corrected and suplemented manuscript with suggestions of Reviewers. We changed the title, supplemented the missing parts. The article has also been supplemented with photos. 

Below I am sending a reply to the review.I hope, that I improved the quality of this manuscript and we have chance to acceptable for publication.

Reviewer 1 (R1)

R 1: Suggest changing the title - it is very difficult to read and may be grammatically incorrect

Suggestion:

Effect of irrigation dose on powdery mildew incidence and root biomass of sessile oaks

Answer (A.): Thank you for your attention. The title was changed as suggested, and the Latin name of the sessile oak was added as suggested by subsequent reviewers. The title now reads: Effect of irrigation dose on powdery mildew incidence and root biomass of sessile oaks (Quercus petraea (Matt.) Liebl.)

R1: Abstract

variation in the extent of irrigation -> irrigation levels

You should state the type of irrigation in the abstract

A.: Corrected and supplemented as suggested.

R1: 30-38 I think it is very important to state that E. alphitoides is not the only Erysiphe species to commonly attack oaks, nor is Erysiphe the only PM genus to attack oaks. Then you need to explicitly state if the common term "oak powdery mildew" is referring only to E. alphitoides within the manuscript or all members of the Erysiphaceae that parasitize oaks. Finally, a sentence somewhere in the manuscript that states whether you believe or took any measures to determine whether the symptoms and signs observed in your study come solely from E. alphitoides.

A.: A paragraph has been added that covers the fungus species causing powdery oak mildew in more detail. See lines 48-62. And this passage already suggests why the authors hold that the powdery mildew of the oak that was tested was caused by E. alphitoides. Then it is explained in more detail in the methodology in lines: 406-408 under the new figure No. 6.

R1: 38 assimilation apparatus -> this term is not familiar to me. This may be my own deficiency, but consider finding a more widely-used synonym

A.: Thank you for your attention. The term assimilation apparatus in the article was replaced by the word leaves.

R1: 48 claim -> this suggests to the reader you may not agree with the finding -> found furthermore, this fact sheet is not a suitable citation - use the reference(s) that the fact sheet cites, not the fact sheet itself

A.: Thank you for your attention . The text has been redrafted and cited the results of the study, not the results of the report.

R1: 53 "dry" -> replace with something more specific (ie they require humidity but not precipitation)

A.: Thank you for your attention changed: Selochnik et al.[23] mentioned an optimum disease development for mean temperature in June around 20°C. Research pointed to the detrimental effects of rain on powdery mildew fungi, by washing-off spores and damaging mycelium at the leaf surface [24]. Sivapalan [25] showed that E. alphitoides for which conidia were able to germinate as well in water as on oak leaves, the ability of the fungus to establish a parasitic relationship with the host strongly decreased with the duration in water.

R1: 56 the oak -> either "the sessile oak" or "oaks"; 61 the oak -> oak and 63 all -> al.

A.: Thank you for your attention. The suggested changes were made.

R1: 70-75 you should mention irrigation methods in addition to irrigation rates, even if this isn't covered by the study

A.: The text was supplemented: Every year the native tree species range of tree stands in Europe, including oak trees, increases. It is associated with more extensive culture of oak trees in nurseries, and this in turn results in a greater risk of outbreaks of powdery mildew [11]. In the nurseries, which are producing seedlings of trees used several irrigation and methods are available. Irrigation can reduce leaching, time of work, use of fertilizer and foliage diseases. Subirrigation systems are commonly used by nursery plant producers. Drip irrigation has been used in areas where water is scarce. Drip irrigation uses up to 70% less water as compared to a flood irrigation system. However, it is expensive compared to other irrigation methods [33]. Drip irrigation was effective in reducing pathogens such as Colletotrichum acutatum (J. H. Simmonds) that cause anthroids in strawberry production [34]. Overhead irrigation is the most common system in outdoor nursery areas. Overhead irrigation is water inefficient (as much as 80%). Major irrigation methods chosen for field nursery irrigation system are overhead sprinklers and microirrigation. The choice of these methods again depends on crop, water type, and quality of plants [33].

R1: 140  you must address the discrepancy in results between years 2015 and 2016 in the discussion. An entire paragraph would not be too long.

A.: This part of the discussion has been supplemented: Consider that it is not only drought that negatively affects plant development. Taking into account the larger amounts of water in 2016, which came from precipitation and lower air temperatures, it can be concluded that the excess of water may contribute to the intensification of infection from powdery oak

In 2015 and 2016, weather conditions may have influenced the existence of differences be-tween soil moisture, the amount of leaf infestation by powdery mildew of oak and the dry mass of oak roots. 2015 was drier compared to 2016, which confirms the existence of differences in soil moisture in 2015 and no differences in 2016. Lower powdery mildew infection of oak may result from the presence of higher current air temperatures in 2015 and lower rainfall. A similar relationship was observed by Markovic et al. [42], who, at a lower air temperature 17-21°C and air humidity of 85-100%, recorded a very high rate of seedling infection.

R1: 200 surface irrigated -> flooding or sprinklers? the delivery method is very important for a powdery mildew study and 200 was the density of the trees in each experimental unit uniform?

A.: In the methodology, we added the following sentences: Density of the trees in each experimental unit was uniform. Plants were surface irriga-tion - overhead sprinklers method (Fig 4a i 4b).

R1: Figure 3 caption replace dose with rate or level

A.: Thank you for your attention. We made the suggested change.

Reviewer 2 Report

This manuscript focuses on the influence of variation in the extent of irrigation on the damage caused by powdery mildew to Quercus petraea growing in a nursery setting. The research topic is important and the results are interesting.

Author Response

Thank you for positive review our manuscript “Influence of irrigation dose on limiting the occurrence of oak powdery mildew and on root biomass of sessile oak trees” to Plants. I hope that we have chance to acceptable for publication, because we took into account the comments of other reviewers and we tried to improve the quality of the presented research results.

Reviewer 3 Report

Dear Authors,

I have reviewed your manuscript "Influence of irrigation dose on limiting the occurrence of oak powdery mildew and on root biomass of sessile oak trees" submitted for publication in the journal Plants. The manuscript is interesting and provides information on an effective strategy to reduce the degree of powdery mildew infestation by controlling the irrigation rate of young sessile oaks. We generally make the association that powdery mildew is favored by dry weather; therefore, the hypothesis studied and the results found are interesting. Looking forward to seeing future research on seedlings kept under controlled conditions.

I suggest some edits that would improve even more the quality of it before publication.

I really recommend including photos showing the different degrees of severity of powdery mildew under different irrigation intensities. Likewise, for seedlings from germination to assessments. This can significantly improve the quality of the manuscript. 

It is necessary to add some references in some sentences in the introduction and material and methods. I have highlighted this in the attached document.

How old were the trees that provided seeds? 
Add spacing between seedlings.
Add the season when mentioning the month of the year.
Add the time of day/average temperature that the irrigations were performed.
The title of table 2 is wrong.
How did you rate the severity of the disease? Add this information to the material and methods.
Line 299: In which portion of the plant (apical, median, basal) were the leaves collected?
Did you notice any difference in disease severity between the blocks? There is a considerable variation between block regarding P, N, K etc.
Have you noticed any influence of the season on the severity of the disease?

For more specific comments, please see the revised manuscript attached.

Kind regards,
Reviewer

Author Response

Dear Reviewer,

Thank you for review our manuscript “Influence of irrigation dose on limiting the occurrence of oak powdery mildew and on root biomass of sessile oak trees” to Plants. We revised manuscript and we added photos. We corrected and suplemented manuscript with suggestions of Reviewers. I hope, that I improved the quality of this manuscript and we have chance to acceptable for publication.

Below I have listed the answers to the Reviewers. 

R3: I really recommend including photos showing the different degrees of severity of powdery mildew under different irrigation intensities. Likewise, for seedlings from germination to assessments. This can significantly improve the quality of the manuscript.

A.: Thank you for your attention. As suggested, we added a table with the results of measuring soil moisture and photos that illustrate: Figure 2. An example of scans of oak leaves taken from plants growing in block A in 2015, infected with powdery mildew of oak, made with a scanner with a resolution of 300 dpi. a) - full dose (100%), b) 75% of the dose, c) 50% of the dose, d) 25% of the dose, Figure 4. The experimental set-up divided into variants of the experiment in Muchów Forest Nursery. 4a -block C and D and working sprinklers, 4b - block A and B and working sprinklers 4c - experiment design - letters (A, B, C and D) stand for block names; 100% - full dose (control), 75% of the rdose, 50% of the dose and 25% of the dose; the dashed line marks the rows where the oak was planted; the desired variant of the experiment is marked with a separate color., 4d - rows in which acorns were planted, the numbers (1-4) are rows.Figure 6. Determine the degree of infestation of scanned leaves with use Compu Eye - Leaf and Symptom Area [61]. Unfortunately, we do not have photos of the roots.

R3: It is necessary to add some references in some sentences in the introduction and material and methods. I have highlighted this in the attached document.

A.: Thanks for your comments. Title changed and authors' addresses added. The abstract includes, inter alia, detailed information on the degree of infection of the leaves by powdery mildew. Quercus petraea has been removed from keywords because the phrase is included in the title. References have been reorganized in the introduction and throughout the manuscript, and missing citations have been added. All abbreviations. replaced with al.

Hypothesis "specifically, we predict that under a lower irrigation rate i) the severity of infection of the leaf blade of the oak will be reduced "aroused the Reviewer's interest. I emphasize that this is only a hypothesis and sometimes it is put against all beliefs. The described experiment was part of a large project on the use of doses irrigation in Polish nurseries for various tree species. When the work began, during the experiment (July 2015) it was noticed that oaks with limited water supply were less infested than those that were watered with a full dose. Hence, we made a hypothesis based on our own observations, but Also based on the idea that rain can wash away the pathogen's mycelium and thus reduce infection, and we used sprinklers.

Specific citations concerning the optimal conditions for powdery mildew of oak were added: The optimum for growth of oak powdery mildew is at 25 ° C and 96% relativehumidity [16, 36]. Light had a significant impact on the growth of mycelium of oak powdery mildew that reached the maximum value in full light [16], although it was not possible to complete it everywhere, because in some publications only general phrases appear, such as in the work: Sucharzewska , E. The Development of Erysiphe alphitoides and E. hypophylla in the urban environment. Acta Mycologica 2013, 44, 109–123, doi: 10.5586 / am.2009.010. and Marçais, B .; Piou, D .; Dezette, D .; Desprez-Loustau, M.L. Can Oak Powdery Mildew Severity Be Explained by Indirect Effects of Climate on the Composition of the Erysiphe Pathogenic Complex? Phytopathology 2017, 107, 570–579, doi: 10.1094 / PHYTO-07-16-0268-R.

Omitted references added in the methodology and conclusions (reference to the Green Deal).

R3: How old were the trees that provided seeds? Add spacing between seedlings.

A.: Stand age - 159 years. Spacing between seeds was 5 cm and between rows was 33.3 cm, Fig. 4d)  Information was added in the text of the article.

R3: Add the season when mentioning the month of the year.

A.: Efforts were made to complete the information as suggested.

R3: Add the time of day/average temperature that the irrigations were performed.

A.: Note added: Plants in the 100% dose variant were watered between 5:00 a.m. and 9:00 a.m., 75% from 5:00 a.m. to 8:00 a.m., 50% from 5:00 a.m. to 7:00 a.m., and 25% from 5:00 a.m. to 6:00 a.m. in subsection: 4.1. Experiment design.

R3: The title of table 2 is wrong.

A.: Table title improved: Results of chemical analyses of soils. Table headers have also been improved.

R3: How did you rate the severity of the disease? Add this information to the material and methods.

A.: The infestation of leaf blades by powdery mildew was expressed as the rate of the affected part (showing etiological symptoms) to the non-infected part (without etiological symptoms) expressed in % (degree of infestation - the severity of the disease, Fig. 6).

R3: Line 299: In which portion of the plant (apical, median, basal) were the leaves collected?

A.: Added note: "Leaves were taken randomly from each part of the plant (apical, median, basal)." in section 4.3. The weighing of dry mass seedlings and calculate leaves area.

R3: Did you notice any difference in disease severity between the blocks? There is a considerable variation between block regarding P, N, K etc. Have you noticed any influence of the season on the severity of the disease?

A.: We did not observe any differences between the blocks. We also did not analyze the impact of the season on the severity of the disease. We assumed that the analysis of the leaf infestation with powdery mildew of oak will be performed in September, when the plants prepare for dormancy in Poland's climatic conditions. During this period, you can also see the full spectrum of powdery mildew infection (primary and secondary infections that occurred throughout the spring and summer season).

I referred to the remaining comments on a regular basis in the text.

Reviewer 4 Report

Some details from material and methods should be moved to results, and discussion should be given according to that parameters evaluated in methods.

Author Response

Thank you for review our manuscript “Influence of irrigation dose on limiting the occurrence of oak powdery mildew and on root biomass of sessile oak trees” to Plants. Below I am sending a reply to the comments. I hope I have explained my position well.

R4: Some details from material and methods should be moved to results, and discussion should be given according to that parameters evaluated in methods.

A.: According to us Table 2. Total quantity of precipitation and irrigation rate for each variant and irrigation dates in years' 2015 and in 2016, VII - July, VIII -August - summer in Poland and Figure 5. The average rainfall [mm] and temperature [ ˚C] and actual rainfall [mm] and temperature [˚C] in Muchów Forest Nursery, Poland (I - January; II - February, III - March etc.) should be included in the methodology, because these results are the basis for determining the dose irrigation. These are the results of measurements and calculations, but necessary to determine the irrigation dose. It is similar with the results of soil analysis. They are only a background to consider. In our opinion, all these elements constitute a fairly coherent whole. Nevertheless, we thank you for this suggestion and hope that our explanation of the location of figures and tables in the methodology is good.

R4: why the author predict athat under lower irrigation rate, the infection will be reduced, when it was showed previously that dry days, inoculation of oak powdery mildew is enhanced, and ....infestation by the oak powdery mildew, is linked to factors such as animal pressure, changes in land use and the diminishing levels of ground water [17] ?

A.: We emphasize that this is only a hypothesis and sometimes it is put against all beliefs. The described experiment was part of a large project on the use of doses irrigation in Polish nurseries for various tree species. When the work began, during the experiment (July 2015) it was noticed that oaks with limited water supply were less infested than those that were watered with a full dose. Hence, we made a hypothesis based on our own observations, but Also based on the idea that rain can wash away the pathogen's mycelium and thus reduce infection, and we used sprinklers (Research pointed to the detrimental effects of rain on powdery mildew fungi, by washing-off spores and damaging mycelium at the leaf surface Sivapalan, A. Effects of Impacting Rain Drops on the Growth and Development of Powdery Mildew Fungi. Plant Pathology 1993, 42, 256–263, doi:https://doi.org/10.1111/j.1365-3059.1993.tb01498.x.).
